# Acetophenone-Based 3,4-Dihydropyrimidine-2(1H)-Thione as Potential Inhibitor of Tyrosinase and Ribonucleotide Reductase: Facile Synthesis, Crystal Structure, In-Vitro and In-Silico Investigations

**DOI:** 10.3390/ijms232113164

**Published:** 2022-10-29

**Authors:** Aamer Saeed, Syeda Abida Ejaz, Aqsa Khalid, Pervaiz Ali Channar, Mubashir Aziz, Qamar Abbas, Tanveer A. Wani, Nawaf A. Alsaif, Mohammed M. Alanazi, Abdullah M. Al-Hossaini, Nojood Altwaijry, Seema Zargar, Muawya Elhadi, Tuncer Hökelek

**Affiliations:** 1Department of Chemistry, Quaid-i-Azam University, Islamabad 45320, Pakistan; 2Department of Pharmaceutical Chemistry, Faculty of Pharmacy, The Islamia University of Bahawalpur, Bahawalpur 63100, Pakistan; 3Department of Basic Sciences and Humanities, Faculty of of Information Science and Humanities, Dawood University of Engineering and Technology, Karachi 74800, Pakistan; 4Department of Biology, College of Science, University of Bahrain, Sakhir Campus, Zallaq 32038, Bahrain; 5Department of Biological Sciences, College of Natural Sciences, Kongju National University, 56 Gongjudehak-Ro, Gongju 314-701, Chungnam, Korea; 6Department of Pharmaceutical Chemistry, College of Pharmacy, King Saud University, P.O. Box 2457, Riyadh 11451, Saudi Arabia; 7Department of Biochemistry, College of Science, King Saud University, P.O. Box 22452, Riyadh 11451, Saudi Arabia; 8Department of Physics, Faculty of Science and Humanities, Ed Dawadmi, Shaqra University, Shaqra 11961, Saudi Arabia; 9Department of Physics, Faculty of Engineering, Hacettepe University, Beytepe, Ankara 06800, Turkey

**Keywords:** thione, dihydropyrimidine, single crystal analysis, DFT, molecular docking

## Abstract

The acetophenone-based 3,4-dihydropyrimidine-2(1H)-thione was synthesized by the reaction of 4-methylpent-3-en-2-one (**1**), 4-acetyl aniline (**2**) and potassium thiocyanate. The spectroscopic analysis including: FTIR, ^1^H-NMR, and single crystal analysis proved the structure of synthesized compound (**4**), with the six-membered nonplanar ring in envelope conformation. In crystal structure, the intermolecular N–H ⋯ S and C–H ⋯ O hydrogen bonds link the molecule in a two-dimensional manner which is parallel to (010) the plane enclosing R_2_^2^ (8) and R_2_^2^ (10) ring motifs. After that, the Hirshfeld surfaces and their related two-dimensional fingerprint plots were used for thorough investigation of intermolecular interactions. According to Hirshfeld surface analysis, the most substantial contributions to the crystal packing are from H ⋯ H (59.5%), H ⋯ S/S ⋯ H (16.1%), and H ⋯ C/C ⋯ H (13.1%) interactions. The electronic properties and stability of the compound were investigated through density functional theory (DFT) studies using B3LYP functional and 6-31G* as a basis set. The compound **4** displayed the high chemical reactivity with chemical softness of 2.48. In comparison to the already reported known tyrosinase inhibitor, the newly synthesized derivatives exhibited almost seven-fold better inhibition of tyrosinase (IC_50_ = 1.97 μM), which was further supported by molecular docking studies. The compound **4** inside the active pocket of ribonucleotide reductase (RNR) exhibited a binding energy of −19.68 kJ/mol, and with mammalian deoxy ribonucleic acid (DNA) it acts as an effective DNA groove binder with a binding energy of −21.32 kJ/mol. The results suggested further exploration of this compound at molecular level to synthesize more potential leads for the treatment of cancer.

## 1. Introduction

Due to the global increase in the number of skin cancer patients, it is desirable to develop new approaches to prevent skin cancer [1]. According to the *World Cancer Report*, 30% of newly diagnosed cancers are skin cancers [2]. This rise in the incidence has been linked to the overexposure of skin to UV rays as a result of ozone depletion [3]. As a result of this overexposure, the malignant cells are identified in the outer layer of the skin, called skin cancer. Among the different types of cancer, melanoma is one of the most dangerous forms of skin cancer, characterized by the proliferation of melanocytes and the accumulation of melanin pigment, resulting in skin pigmentation and discolouration as well as tumour growth. Moreover, it appears that the elevated level of tyrosinase enzyme significantly contributes to melanin synthesis and accumulation in melanocytes. Like other cancers melanoma responds well to early detection and treatment, but in late stages it can rapidly spread throughout the body via the lymphatic system and bloodstream. The vast majority of cytotoxic drugs currently used in the cancer treatment are extremely hazardous to a variety of tissues, including the digestive tract, bone marrow, heart, lungs, kidney, and brain, and lactogenic failure of these organs is a common cause of cancer-related mortality [4]. Due to a well-known phenomenon known as “chemoresistance,” melanomas are difficult to treat with chemotherapy. It appears that the expression of surviving molecules in cells causes drug resistance, leaving patients with few treatment options [5]. In addition to the tyrosinase, the ribonucleotide reductase (RNR) is a multi-subunit enzyme that catalyzes the rate-limiting step in deoxynucleoside triphosphate (dNTP) synthesis, specifically the conversion of ribonucleoside diphosphates to deoxyribonucleoside diphosphates [6]. This enzyme is considered as a key marker for the determination of normal physiology of the cell replication. The aberrant expression of this RNR has long been considered an appealing therapeutic target for a variety of proliferative disorders, including cancer [7]. Therefore, the exploration of tyrosinase inhibitors along with the inhibitor of RNR is considered as an emerging field for the treatment of skin cancer to overcome chemoresistance and side effects of already available drugs.

Among different heterocyclic compounds, dihydropyrimdin-2(1H)-thiones [8] are heterocyclic compounds that include a pyrimidine ring system [9]. These compounds are often produced by variants of notable three-component [10] Biginelli [11] reaction. The dihydropyrimidines and Dihydropyrimidine-2-thiones [12] have various biological activities, ref. [13] including antiviral, antibacterial, hypotensive [14], anti-inflammatory [15] anti-oxidant properties [16] and act as calcium channel blocker [17,18]. These compounds can be utilized as an evaluator of calcium channels [19] and multiple reflections of drug resistance [20]. The ring structure of pyrimidine is present in several biomolecules such as nutrients and nucleic acids, and assumes a significant part in different biological processes [21]. Since the discovery of monastrol as low molecular weight anticancer agent, a radical increase in the synthesis of dihydropyrimidine derivatives has been observed, as shown in Figure 1 [15,21,22,23].

In continuation of our previously reported work on these compounds, here we present the synthesis of acetophenone-based 3,4-dihydropyrimidine-2(1H)-thione molecule and its potential to inhibit tyrosinase as well as RNR. Moreover, in the present era, computational chemists investigate the bond length, number of torsion, bond angle, harmonic frequencies and chemical changes at molecular level, using both theoretical and experimental data. The in-vitro results were confirmed by in-silico techniques that included molecular docking investigations, density functional theory calculations and Hirshfeld analysis. The DFTs are a crucial tool for computational chemists in determining the comprehensive structural characteristics of the produced crystal, and Hirshfeld analysis provides the substantial information about intermolecular interactions of these compounds. In addition, DNA-binding docking studies were performed to evaluate the crystal as a powerful DNA groove binder. The results of experimental data were found in correlation with the theoretical data, suggesting this acetophenone-based 3,4-dihydropyrimidine-2(1H)-thione as a lead molecule for the synthesis of potential derivatives for the treatment of skin cancer and other associated problems.

## 2. Results and Discussion

### 2.1. Synthesis

The 1-Aryl-4,4,6-trimethyl-3,4-dihydropyrimidine-2-(1H)-thiones is an exceptional subtype of 3,4-dihydropyrimidine-2-thiones and target compound (**4**), which can be synthesized with great ease. Acetone was used as both the solvent and reagent, and self-condensation of acetone afforded the key intermediate 4-methylpent-3-en-2-one **5** (Figure 1) which reacted with KSCN and 4-acetyl aniline (**1**) to afford the title molecule (**4**) in excellent yield 95%. The purification and isolation of the product was performed by recrystallization using ethanol as solvent. Single crystal analysis revealed the structure of compound (**4**) and was confirmed by Fourier transform infrared (FT-IR) and ^1^H-NMR.

The thioamide moiety had a characteristic stretching vibration in the range 1733 cm^−1^, while the olefinic bond stretch was between 1597 and 1696 cm^−1^. The singlet resonance of the thioamide proton (NH) occurred at *δ* 8.07 ppm in the ^1^H-NMR spectrum, while aromatic protons resonated around *δ* 8.06–7.39 ppm. Two singlets of three protons corresponding to Csp^2^–CH_3_ appeared at *δ* 5.03 ppm, whilst the singlet for acetyl function appeared at *δ* 2.87 ppm. The germinal dimethyl of pyrimidine ring showed a singlet at *δ* 1.52 pm.

### 2.2. X-ray Analysis

The (N1/N2/C1—C4) ring is in envelope conformation in the molecule of the title compound (Figure 2), with puckering parameters of Q_T_ = 0.0853 (25) Å, θ_2_ = 59.51 (17)° and φ_2_ = 323 (2)°, where the C2 atom is at the flap position and is 0.1285 (26) Å away from the best plane of the other atoms. The titled compound is composed of pyrimidine and phenyl rings.

The interatomic distances obtained from experimental X-ray diffraction were in accordance with theoretical bond lengths obtained from DFT/B3LYP/6-31G*. Table 1 below summarizes interatomic distances. The bond lengths between carbon and hydrogen atoms are average 2.80 Å, whereas C-C bond length is 3.365 Å. When these values were compared with theoretical calculations, there was no significant difference between theoretical and experimentally obtained bond lengths. Then the correlation coefficient value was calculated for bond lengths, which was 0.9912.

Intermolecular N–H⋯S and C–H⋯O hydrogen bonds (Figure 3) in the crystal link the molecules, encapsulating R_2_^2^ (8) and R_2_^2^ (10) ring motifs into a two-dimensional architecture parallel to (010) plane of the crystal [24]. The Table 2 depicts all the experimental values and all are found within the normal range [25,26]. The X-ray diffraction data revealed that compound crystallizes in triclinic system. The unit cells dimensions were a = 6.8262 (4) Å, b = 10.7036 (9) Å, c = 11.1396 (7) Å and V = 758.37 (Å^3^).

The interatomic hydrogen bond distances were calculated for title compound and presented in Table 3. It was observed that there was three *D*—H⋯*A* (edge to face) interactions. Elaboration of these interactions is provided in Table 2. The N1 present in pyrimidine ring act as hydrogen bond donor to S1 of a molecule. The C3 of phenyl ring acts as hydrogen bond donor via H3 to O1 of the molecule. Similarly, C7 was also acting as hydrogen bond donor via H7 to O1 of molecule.

### 2.3. Hirshfeld Surface Analysis

The Hirshfeld surface analysis was performed with the title compound using standard analysis protocol [27]. The results of HS analysis are mentioned in the Figure 4, Figure 5 and Figure 6. All other data relevant to Hirshfeld surface (HS) analysis along with significance of the study are provided in the Appendix A for consultation see [28,29].

The different-coloured regions of HS (Hirshfeld surface) analysis revealed different electrostatic potential and contact distances. Particularly, red-coloured regions exhibit shorter contacts and negative electrostatic potential with hydrogen bond acceptor profile. In contrast, blue-coloured regions depict relatively longer contacts and positive electrostatic potential with hydrogen donor profile. Different electrostatic regions are depicted in Figure 4 and Figure 6.

The fingerprint plots are represented in Figure 5. Figure 5a represents collective fingerprint plots, whereas Figure 5b–j represents fingerprint plots explained into H ⋯ H, H ⋯ S/S ⋯ H, H ⋯ C/C ⋯ H, H ⋯ O/O⋯ H, S ⋯ C/C ⋯ S, O ⋯ C/C ⋯ O, H ⋯ N/N ⋯ H, S ⋯ N/N contacts. The most dominant contacts were observed to be H ⋯ H, which accounts for 59.5% packing of crystal. The dominancy of H ⋯ H contacts is depicted in Figure 5b, which showed scattered points due to high density of hydrogen contacts with the tip at de = di = 1.17 Å. Further detailed discussion is provided in the Appendix A.

### 2.4. Density Functional Theory Studies

The geometry of compound **4** was optimized using B3LYP functional correlation and the basis set as 6-31G*. Structure was optimized to the lowest energy gradient and no imaginary frequencies were obtained, which demonstrated that obtained geometries were true local minima. Optimization energy of compound **4** was −1146.040 hartree and demonstrated polarizability value of 108.56 (a.u). Dipole moment is a critical parameter to determine the polarity of compound. It can arise in ionic and covalent bond due to electronegativity differences. Dipole moment value of 4.043 Debye exhibit significant difference in electronegativity and polarity among individual atoms of title compound [30]. Geometric parameters of the compound are tabulated in Table 4.

The frontier molecular orbitals (FMOs) are predictive of compounds stability and reactivity. There are two types of FMOs. HOMO is the most occupied molecule orbital, while LUMO is the least populated molecular orbital. The nucleophilic component is HOMO. The electrophilic component is LUMO. HOMO always donates an electron. In contrast, LUMO always accepts an electron. However, HOMO is the outermost orbital while LUMO is the innermost orbital. The distribution of FMOs for the titled compound was calculated at DFT/B3LYP level of theory. The theoretical calculation demonstrated that titled compound had 84 occupied orbitals. It was observed that LUMO orbitals of compound **4** were confined over the benzene ring, which suggested that electrophilic character is due to presence of benzene ring, whereas HOMO orbitals were majorly confined to the pyrimidine ring. The two nitrogen atoms present in the pyrimidine ring are acting as electron pair donor and nucleophilic character is dominated due to presence of these two nitrogen atoms. The HOMO and LUMO energies were calculated to be −0.197 eV and 0.207 eV, respectively. These energy values (HOMO and LUMO) were utilized to determine the electronegativity, electron affinity, chemical reactivity, softness and hardness of the titled compound. The FMOs orbital of compound **4** is shown in Figure 7 given below.

The energy gap between LUMO and HOMO predicts the reactivity and kinetic stability of compounds. Compound **4** was detected to be highly reactive because of small LUMO/HOMO energy gap (0.404 eV). Another important parameter to determine the reactivity of the compound is chemical hardness and softness of the molecules. The compound **4** showed good value for chemical softness (2.48 eV), which showed reactive behaviour of the compound. The energetic parameters are tabulated in Table 5.

### 2.5. Mushroom Tyrosinase Inhibition Assay

Synthesized crystal was evaluated for their pharmaceutical potentials, In vitro results showed that crystal was highly potent inhibitor of tyrosinase with IC_50_ 1.977 µM as compared to the reference drug kojic acid IC_50_ 15.79 µM. Crystal was found to be 7-fold more potent as compared to reference as tyrosinase inhibitors (Figure 8A,B).

### 2.6. Molecular Docking Approach

The binding orientation and nature of molecular interactions produced by the title compound within the active pocket of the targeted protein was determined by molecular docking approach. Molecular docking is an efficient technique for generating the reliable docking poses and provides significant insight into the inhibitory potential of the compound. In the current study, The MOE software was utilized for docking of compound **4** inside active pockets of tyrosinase, RNR protein and DNA.

Tyrosinase is involved in catalysing the oxidation process by molecular oxygen. These reactions are responsible for malignancies in animals and degradation of fruits and vegetables, so it is important to target tyrosinase enzyme for the alleviation of several oxidation-related malignancies. The targeted protein was retrieved from a protein data bank using PDB ID: 4OUA (resolution; 2.76 Å). The top ranked conformation was selected on the basis of docking score and binding affinity and visualized using PyMOL software [31]. Compound **4** showed good hydrophobic and hydrophilic interactions with mushroom tyrosinase. The docking score was observed to be −19.70 kJ/mol. The complex was stabilized by two hydrogen bonds with the bond lengths of 3.3 and 4 Å, respectively. Furthermore, the hydrophobic interactions were observed with THR114, VAL115, LYS118, GLU119 and HIS116 amino acid residues. These molecular interactions were contributing toward the formation of stable protein–ligand complex. The putative 2D and 3D binding mode of the compound **4** is shown in Figure 9, in which compound **4** is represented as yellow lines whereas the amino acid residues of mushroom tyrosinase are represented as blue lines.

Ribonucleotide reductase (RNR) catalyzes the reduction of ribonucleotides to their respective deoxy ribonucleotides, which aids in the de-novo synthesis of DNA precursors. RNR in higher organisms is made up of two dimers i.e., R1 and R2. R1 is a bigger dimer with binding and allosteric sites for substrate binding. It also has an essential cysteine residue that facilitates radicle transfer between the R1 and R2 subunits. Tyrosyl [32] binding facilitates radical transport. RNR’s crystallographic structure is shown in Figure 10.

R1 subunit of RNR protein contains important amino acid residues, which are involved in activation of protein. The amino acid residues of active site were selected after extensive literature review. The site finder utility of MOE was used to select important residues of active site, which were selected were as follows: ARG695, LYS613, ARG695, TYR693, LEU699, GLY590, HIS603, SER600, PHE666, GLU610, ARG612 PRO588, TYR692 and TYR594. These amino acid residues were selected and docked with compound **4**. The most stable conformation based on docking score was selected and analysed for further analysis. Figure 11 shows most probable binding interactions of compound **4** with in activation loop of RNR.

It was seen that docked conformation of compound **4** showed important bonded and non-bonded interactions with amino acid residues of active site. The amino acid residues involved in non-bonding interactions were as follows: GLU610, ILE611, LUS613, HIS603, TYR594, PHE666, THR664, PRO588, TYR693, TYR692, VAL606 and ARG695. Although, ARG612 was only involved in the formation of hydrogen bonding with oxygen atoms of the compound. The bond length of hydrogen bond was 2.9 Å. Hydrogen bonding was a major electrostatic attraction that was responsible for formation of the stable protein–ligand complex. Non-bonding interactions were comprised of alkyl-alkyl, π-alkyl and van der Waal attractions, which were also contributing toward stable complex formation. Moreover, the compound produced a docking score of −19.68 kJ/mol. These potential molecular interactions were testimony for the inhibiting potential of (4) against DNA synthesis. RNR plays an important role in DNA synthesis; hence inhibition of RNR is critical in alleviating the cancer malignancies.

In-silico DNA docking studies were carried out using MOE docking software. The DNA docking studies were performed using DNA structure (PDB ID: 127D), co-crystallized with Hoechst. These studies were performed to investigate the compound **4** as a major groove binder. The docking protocol was validated by re-docking of co-crystal ligand i.e., Hoechst (minor groove binder) with DNA structure. It was observed that re-docking protocol showed RMSD value less than 2 Å. The compound **4** was docked into DNA groove which exhibited strong molecular interactions with important bases (mainly purines) of DNA. It was observed that compound **4** showed interactions with AAT sequence of DNA groove. These studies revealed compound **4** as a potent groove binder. The compound **4** produced stable conformations with the docking score of −21.32 kJ/mol. Figure 12 in-silico studies supported the experimental data as compound **4** formed stable interactions with both RNR and DNA.

### 2.7. Molecular Dynamic Simulation

The nature of molecular interactions identified by the molecular docking method was further investigated using molecular dynamic simulation studies. MD simulations are a potential method for studying the stability of protein–ligand complexes. To measure the stability of a protein–ligand complex, various analytic matrices such as RMSD and RMSF are used. The root mean square deviation (RMSD) is a measure of the average displacement of atoms and molecules with respect to a reference frame. The average RMSD of protein tyrosinase was 2.7 angstrom and the complex average was found 3.3 angstrom, which is an acceptable value. The in-vitro and in-silico inhibitory potential of compound **4** against mushroom tyrosinase enzyme was studied in the current investigation. Both investigations show considerable inhibition of the tyrosinase enzyme, which is validated by MD simulation studies. Figure 13 and Figure 14 depicted the RMSD and RMSF patterns of a protein (mushroom tyrosinase) and a protein–ligand complex, respectively.

## 3. Materials and Methods

### 3.1. Experimental

Melting points were evaluated using the validated digital Gallenkamp (SANYO) model MPD BM 3 equipment. The FTIR spectra were recorded using an FTS 3000 MX spectrophotometer. ^1^H NMR analysis was done in acetone-d_6_ solutions at 300 MHz and 75.4 MHz, respectively, using a Bruker AM-300 spectrophotometer. The chemical shifts were measured in ppm units and coupling constants (*J*) in hertz.

### 3.2. Synthesis of 1-(4-(4,4,6-Trimethyl-2-thioxo-3,4-dihydropyrimidin-1(2H)-yl)phenyl) Ethanone (4)

An amount of 4-acetyl aniline (1) (0.141 g, 1.0 mmol) was added in portion to a mixture of potassium thiocyanate (3) (0.097 g, 1.0 mmol) in acetone (2) containing one drop of HCl at room temperature. The reaction mixture was heated at 50–60 °C for 3 h. The progress of reaction was monitored using thin layer chromatography (TLC) method by observing the complete consumption of aniline (1). On completion, the reaction mixture was cooled to room temperature and poured into ice water. The precipitated compound was filtered, dried and recrystallized from ethanol by slow evaporation at room temperature to afford the purified 1-(4-(4,4,6-trimethyl-2-thioxo-3,4-dihydropyrimidin-1(2H)-yl)phenyl) ethanone (**4**) in excellent yield.



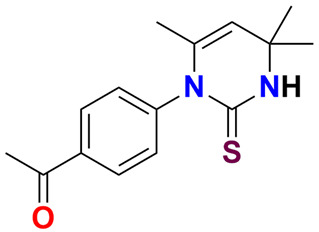



Yield: 85%, *R*f: 0.43 (EtOAc: Petroleum ether 1:9); M.P: 133–134 °C; IR: 3210 (N-H), 3054 cm^−1^ (=C–H stretch), 1696 cm^−1^ (S–NH), 1598 cm^−1^ (C=C), 1260 cm^−1^ (C–N); ^1^H-NMR (300 MHz, acetone-d_6_): *δ* 8.07 (s, 1H, NH), 8.06 (d, 2H, *J* = 8.7 Hz, Ar-H), 7.39 (d, 2H, *J* = 8.7 Hz, Ar-H), 5.03 (s, 1H, Csp^2^–H), 2.87 (s, 3H, COCH_3_), 2.63 (s, 3H, Csp^2^–CH_3_), 1.52 (s, 6H, 2 × CH_3_); ^13^C-NMR (75 MHz, acetone-d6): δ 195.9 (C=O), 170.2 (CSNH), 145.1, 142.1, 128.9, 122.8, (Ar-C) 134.6 (−C=C–), 108.2 (−H–C=C–), 60.0 ((CH3)2–), 40.0 ((CH3)2–C), 25.7 (COCH3) 17.7 (CH3) δ 185.7 (CSNH), 136.0 (−C=C–), 135.1, 130.0, 127.4, 125.3, 121.7, 121.2, 108.2 (−H–C=C–), 58.9 ((CH3)2–), 55.9 (Ar–OCH3), 35.0 ((CH3)2–C), 32.0 (−C–CH3).

### 3.3. X-ray Crystal Structure and Refinement

The crystallographic results of the title compound were acquired using a Rigaku Oxford diffraction Eos, Gemini diffractometer using Cu Kα radiation (λ = 1.5418 Å). SHELX programme packages [33] were utilized to solve and refine the structure using the multi-scan absorption correction [34] applied data, and ORTEP-3 [35] programme was employed in drawings. The locations of hydrogen atoms were calculated geometrically at distances of 0.88 (for NH), 0.95 (for CH), and 0.98 (for CH_3_), and then refined using a riding model by applying the constraints of Uiso (H) = k × Ueq (C, N), where k = 1.2 for NH and CH hydrogens and k = 1.5 for CH_3_ hydrogens. The Cambridge Crystallographic Data Centre (CCDC) has deposited crystallographic data for the structure described here as Supporting Information, CCDC No. 2195791. Copies of the data can be obtained through application to CCDC, 12 Union Road, Cambridge CB2 1EZ, UK. (fax: +44 1223 336033 or e-mail: deposit@ccdc.cam.ac.uk or at http://www.ccdc.cam.ac.uk (accessed on 20 July 2022)).

### 3.4. Hirshfeld Surface Studies

Hirshfeld surface (HS) analysis is the most reliable method for analyzing interactions present in crystal structures [36,37]. The intermolecular interactions in the crystal structure of the (4) were identified using Crystal Explorer 17.5 [38]. On the inside and outside of the surface, the Hirshfeld surface distance from the nearest nucleus was measured and represented by d_i_ and d_e_, respectively, while D_norm_ was utilized to represent a normal contact distance. D_norm_ has been represented by the colours white, blue and red. The input file is in CIF format and can be found in the Appendix A.

### 3.5. Density Functional Theory Studies

The ground state geometry of the selected compounds was optimized using density functional theory calculations, which is a reliable computational approach. DFT calculations [39] were used to optimize the ground state geometries, with B3LYP as functional correlation and 6-31G* as a basis set [39,40]. It is important to determine the electronic properties of the compound as chemical reactivity and stability of compound depend on electron density, DFT is a theoretical approach which provide precise assumptions on the electronic behaviour of the compound [41,42]. The initial structures were completely optimized using Guassian 09W software [43]. The energy minimization was done by omitting the imaginary frequencies in a harmonic vibrational analysis [44]. Visualization was carried out using Guass view 6.0 [45].

### 3.6. Mushroom Tyrosinase Inhibition Activity

The inhibition of mushroom tyrosinase (Sigma Chemical, USA) was carried out according to our previously published methods [46]. The Appendix A contains the complete protocol.

### 3.7. Molecular Docking Methodology

Molecular Operating Environment 2015.10 (MOE) [47] was used to conduct the docking experiments. The crystal structure of ribonucleotide reductase and mushroom tyrosinase (PDB ID: 2BQ1 and 4OUA, respectively, resolution: 3.99 and 2.76 Å) were taken from the protein data bank (https://www.rcsb.org/ (accessed on 25 July 2022)) [48]. MOE’s protein preparation wizard was used to rectify the protein for docking studies. Het atoms and water molecules were eliminated first, followed by 3D protonation at pH = 7 and energy minimization to the lowest gradient. The MMFF94x force field was used to introduce partial charges and polar hydrogens. The amino acid residues of active pockets were selected using MOE’s site finder programme after the protein was prepared. The Chemdraw Ultra 12.0 was used to create the structure of compound **4**, which was then docked into the active pocket of the targeted proteins using MOE’s default parameters. Placement was done with a triangular matcher, and refinement was adjusted to induce fit [49]. For compound **4**, the scoring function London dG was employed, and 100 poses [50] were generated.

For investigation of binding orientation of compound **4** within DNA, docking was performed using MOE. Crystallographic DNA structure was retrieved from protein data bank (PDB ID: 127D). The DNA dodecamer d (CGCGAATTCGCG) was in complex with minor groove binding drug Hoechst 33258. The DNA has specific nucleophilic bases sites (N7, N3 of purines) and O6 position of guanine base pairs present in minor grooves of DNA. These sites are majorly responsible for the DNA functioning and carcinogenic conversion of the cell. The co-crystal ligand Hoechst 33258 lies in the narrow minor groove of B-DNA in the AATT region. During the study, the dimensions of Hoechst 33258 were used and searched for base pairs of active sites using the site finder utility of MOE software. After identification of an active site, the selected compound **4** was docked into the minor groove of DNA and the binding mode was investigated. A total of 100 poses was generated, and the best protein–ligand conformation was selected on the basis of highest docking score and predicted binding affinity (*ki*) for evaluation of molecular interactions.

In order to validate the docking protocol and algorithm implementation for the generation of reliable binding interactions, re-docking of co-crystal ligand was performed, and molecular docking protocol was accessed for reproducibility of binding pose in comparison to native co-crystal ligand complex. The root mean square deviation (RMSD) of less than 2 angstrom for a particular pose was considered as validated. In addition, MOE output results also provide RMSD of each pose which further validate the docking protocol [51,52].

### 3.8. Molecular Dynamics Simulation Studies

The MD simulation Studies were conducted according to our previously reported studies [53]. The detail protocol is given in the Appendix A.

## 4. Conclusions

The title compound, 3,4-dihydropyrimidine-2(1H)-thione (4) was synthesized and validated experimentally and theoretically as the potential inhibitor of tyrosinase and RNR protein. The molecular geometry parameters and crystal structure of the compound were retrieved. The crystallographic studies revealed that the molecular configuration is stabilized by intermolecular interactions, most importantly N–H⋯S and C–H⋯O hydrogen bonds link the molecules, enclosing R_2_^2^ (8) and R_2_^2^ (10) ring motifs. The most important contributions for the crystal packing are from H ⋯ H (59.5%), H ⋯ S/S ⋯ H (16.1%) and H ⋯ C/C ⋯ H (13.1%) interactions, which indicate the dominant nature of van der Waals interactions. Experimental findings were supported by theoretical studies which revealed the reactive properties and predict compound **4** as a potent tyrosinase inhibitor and DNA groove binder. On the basis of findings, it can be concluded that compound **4** can act as a multi-target inhibitor with promising DNA groove binder properties that can efficiently block the DNA proliferation and cell growth.

## Data Availability

Not Applicable.

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
