# Peer review of "Acetophenone-Based 3,4-Dihydropyrimidine-2(1H)-Thione as Potential Inhibitor of Tyrosinase and Ribonucleotide Reductase: Facile Synthesis, Crystal Structure, In-Vitro and In-Silico Investigations"

_ijms, 2022, doi:10.3390/ijms232113164_

Round 1
Reviewer 1 Report
The manuscript submitted by Saeed et al. entitled “Acetophenone-based 3,4-dihydropyrimidine-2(1H)-thione as 2 Potential Inhibitor of tyrosinase and Ribonucleotide eductase: 3 Facile synthesis, Crystal Structure, In-vitro and In-silico 4 Investigations” describes the synthesis of 3,4-dihydropyrimidine-2(1H)-thione .The introduction part is written fairly with approaches involving in the synthesis of dihydroyrimidine thione and about the statistics and cause of skin cancer and therapeutics development against the same like tyrosinase/ribonucleotide reductase inhibitors. Data representation and data quality and spectral qualities are good. Moreover, detailed X-ray analysis, in-vitro testing were done and confirmed by in-silico techniques that included molecular docking investigations, density functional theory calculations and Hirshfeld analysis. In addition, DNA-binding docking studies were performed to evaluate the crystal as a powerfulDNA groove binder.The following points should be clarified
1) The 13 C NMR data should be written up with mention the type of carbon like –C, -CH, - CH 2 and –CH 3 against each value.
2) The most important point is that all the experiments to results are based on a single synthetic compound. I would prefer the synthesis of some more analogues based on the same strategy followed by docking and bio-evaluation for all of the compounds and then conclusion regarding the lead compound/compounds.
In my opinion the manuscript is not suitable for publication in an esteemed journal like “International Journal of Molecular Sciences” in its current form. After the inclusion of above mentioned major work it can be resubmitted.
Author Response
Reviewer 1
The following points should be clarified
- The 13 C NMR data should be written up with mention the type of carbon like –C, -CH, - CH 2 and –CH 3 against each value.
Response: Suggested changes has been incorporated in main text
2) The most important point is that all the experiments to results are based on a single synthetic compound. I would prefer the synthesis of some more analogues based on the same strategy followed by docking and bio-evaluation for all of the compounds and then conclusion regarding the lead compound/compounds.
Response: We are thankful to the reviewer for this fruitful suggestion. Specific drugs for the treatment of skin cancer are very few. This crystal structure is new and depending upon the results it is found as the best tyrosinase inhibitor and RNR inhibitor. We want to publish this compound first as this pure compound can act as lead compounds that can be used for skin cancer treatment. Moreover, we are in the process to synthesize the derivatives based on the results, and in a future publication, we will compare the result. The authors are thankful in advance to the reviewer for understanding our situation and we do hope that reviewer will allow the publishing article with one pure compound

Reviewer 2 Report
The manuscript may be revised according to the following comments:
1. In Introduction section: abbreviation dNTP should be spell out.
2. Please, change the colour of sulfur atom on figures and schemes - it it almost invisible.
3. Why on fig. 8 the tyrosinase activity increases with the crystal and kojic acid concentration growth? It seems like both compounds act as activator of tyrosinase, not as inhibitors.
4. Please, provide 13C NMR description of compound 4 in Experimental section
Author Response
The manuscript may be revised according to the following comments:
- In Introduction section: abbreviation dNTP should be spell out.
Response: dNTP is abbreviated as deoxynucleoside triphosphate which has been incorporated in main text.
- Please, change the colour of sulfur atom on figures and schemes - it it almost invisible.
Response: Correction has been made as suggested.
- Why on fig. 8 the tyrosinase activity increases with the crystal and kojic acid concentration growth? It seems like both compounds act as activator of tyrosinase, not as inhibitors.
Response: In figure 8, percentage inhibition of tyrosinase is plotted against log concentration. it is mistakenly written as tyrosinase activity which has been corrected. Kojic acid is standard inhibitor of tyrosinase. Reference is provided below;
doi: 10.3390/ijms10062440
- Please, provide 13C NMR description of compound 4 in Experimental section
Response: Description of 13C NMR is incorporated in the experimental section.

Round 2
Reviewer 1 Report
Based on the response received regarding the synthesis of analogs later based on this result, I think the manuscript may be accepted subject to editorial corrections.